# RISE: Rolling-Inspired Scheduling for Emergency Tasks by Heterogeneous UAVs

Bowen Fei [1], Daqian Liu [1], Weidong Bao [1,*], Xiaomin Zhu [1,2] and Mingyin Zou [1]

1 College of Systems Engineering, National University of Defense Technology, Changsha 410073, China
2 Center for Assessment and Demonstration, Academy of Military Sciences, Beijing 100091, China
* Correspondence: wdbao@nudt.edu.cn

**Abstract:** The multiple unmanned aerial vehicles (UAVs) system is highly sought after in the fields of emergency rescue and intelligent transportation because of its strong perception and extensive coverage. Formulating a reasonable task scheduling scheme is essential to raising the task execution efficiency of the system. However, the dynamics of task arrival and the heterogeneity of UAV performance make it more difficult for multiple UAVs to complete the tasks. To address these issues, this paper focuses on the multi-UAV scheduling problem and proposes a method of rolling-inspired scheduling for emergency tasks by heterogeneous UAVs (RISE). In order to ensure that emergency tasks can be allocated to UAVs in a real-time manner, a task grouping strategy based on a density peaks (DP) clustering algorithm is designed, which can quickly select UAVs with matching performance for the tasks arriving at the system. Furthermore, an optimization model with multiple constraints is constructed, which takes the task profit and UAV flight cost as the objective function. Next, we devise a rolling-based optimization mechanism to ensure that the tasks with shorter deadlines are executed first while maximizing the objective function, so as to obtain the optimal task execution order for each UAV. We conduct several groups of simulation experiments, and extensive experimental results illustrate that the number of tasks successfully scheduled and the utilization rate of UAVs by RISE are superior to other comparison methods, and it also has the fastest running time. It further proves that RISE has the capability to improve the completion rate of emergency tasks and reduce the flight cost of multiple UAVs.

**Keywords:** multi-UAV scheduling; emergency tasks; rolling-based optimization mechanism; task profit; UAV flight cost

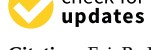



## 1. Introduction

Unmanned aerial vehicles (UAVs) offer the advantages of being lightweight and low cost compared with manned aircraft. They can carry out specified tasks according to the load type via remote control or independent planning without the need for operators. For these reasons, UAVs are frequently utilized as an alternative to performing difficult and hazardous activities [1,2]. The task execution mode has gradually shifted from single UAV to multi-UAV cooperation with the ongoing advancement of relevant technologies and the diversification of application scenarios [3]. A single UAV cannot complete complex tasks independently due to its restricted load and performance, especially in the face of dynamic changes of the task environment [4]. As multiple UAVs can be used simultaneously to perform tasks in a coordinated and robust manner amid a complex environment, multiple UAV systems have received extensive attention in the fields of emergency rescue, intelligent transportation, and logistics distribution in recent years [5,6].

A multi-UAV system has clear advantages in both space and time of task execution. For one thing, a group of UAVs can complete multiple tasks simultaneously in different geographical locations [7]. Additionally, these UAVs can flexibly cope with the changing environment. When one of the UAVs is unable to perform tasks due to interference or

breakdown, its incomplete tasks can be reassigned so that the system is less affected and has better fault tolerance [8]. Moreover, the multi-UAV system has presented the distinct characteristics of various types and wide applications with the rapid development of UAV technology. The UAVs in the system are quite different in many aspects, such as size, weight, mileage, flight time, and flight speed [9,10]. Hence, the system composed of a group of heterogeneous UAVs is widely used, which can better adapt to scenarios involving diverse task types and unknown threat environments. It is worth noting that the key to ensuring that the multi-UAV system can accomplish the tasks efficiently is formulating a reasonable task planning scheme and clarifying the division of labor of each UAV. Task scheduling, a crucial component of task planning, can assist multiple UAVs to obtain higher benefits on the premise of completing tasks [11]. However, multi-UAV task scheduling is a NP-hard problem, and numerous studies focus on modeling and solving this problem. Some work [12,13] models multi-UAV task scheduling as the traveling salesman problem (TSP) or the vehicle routing problem (VRP). On this basis, the variants of the above models are also proposed successively by considering the constraints such as task completion time and UAV load capacity [14,15], and they have better adaptability to the environments. These methods are suitable for the situation of fewer tasks and stronger UAV capability and have the advantages of simple principles and low complexity. However, they need to simplify the relevant constraints or task elements to seek model standardization in the modeling process. In addition, the choice of the proper solution algorithm for the established model is another significant issue in the research on multi-UAV task scheduling [16], which is determined by the task scenario, number of tasks, performance of UAVs, and communication mode, among other factors.

*Motivation.* Multi-UAV task scheduling is a process in which a mission is decomposed into some number of tasks, and these tasks are assigned to UAVs in a multi-UAV system to complete them, with the goal of achieving the optimal or suboptimal performance of a certain criterion function [17]. Nevertheless, this process is limited by constraints such as task execution time and UAV performance. The majority of current research uses the conventional model and solution algorithm when addressing the multi-UAV task scheduling problem. These approaches struggle to adapt to the complex and changeable task environment. As a result, the tasks' features in the actual scenario still need to be carefully examined. In particular, there are two issues that need to be taken into consideration in the design of the multi-UAV task scheduling scheme:

- The rationality of task scheduling. The tasks that will be assigned should be compatible with the UAV performance. For example, the UAV that carries materials can carry out the tasks of distribution and rescue. However, most of the existing task scheduling schemes fail to establish a certain mapping relationship between tasks and UAVs, which prevents the performance of UAVs from meeting the requirements of the tasks that are assigned to them, ultimately leading to task rejection or execution delays that have an impact on the multi-UAV system's overall efficiency.
- The adaptability of task scheduling. The arrival of emergency tasks and the depletion of UAV energy frequently occur during the task execution of the multi-UAV system. Some methods build task scheduling models based on the known task environment but lack dynamic scheduling mechanisms that can adapt to the changes in the environment. Once an emergency occurs, the established scheme cannot be applied to the changed scenario, which easily leads to the problems of low task completion rate and low utilization rate of UAVs.

In summary, there is an urgent need for an emergency task scheduling method with a real-time and rapid response capability that can effectively save unmanned resources and improve task completion efficiency. This kind of method can quickly generate the task UAV matching mechanism after the attributes of the tasks and UAVs change, and it can modify the original task execution sequence and scheduling scheme in a real-time manner to maximize the benefits of task execution. Aiming at the above issues, we propose

a method of <u>R</u>olling-<u>I</u>nspired <u>S</u>cheduling for <u>E</u>mergency tasks by heterogeneous UAVs (RISE). The main contributions of our work are as follows:

**Contributions**: The major contributions of this paper are summarized as follows:

- We construct an optimal scheduling framework for emergency tasks, which is mainly composed of two phases: task clustering and optimization scheduling. It can rapidly generate scheduling schemes for tasks that arrive in the system dynamically.
- We devise a task clustering strategy inspired from a density peaks (DP)-based clustering method, which realizes the reasonable grouping of dynamic tasks by constructing a distance matrix based on multiple task attributes.
- We present a dynamic task scheduling model for heterogeneous UAVs. We create an objective function that takes the value and completion probability as the task profit, and it also takes the safety risk and energy consumption as the UAV flight cost. Furthermore, a task allocation strategy based on the rolling optimization idea is designed to improve the execution efficiency of emergency tasks.

The remainder of this paper is organized as follows. Section 2 introduces the related work. The system framework and problem statement are depicted in Section 3. Section 4 provides a thorough explanation of the modeling and formulation. The proposed task scheduling method is introduced in Section 5, followed by the simulation experiments and performance analyses in Section 6. Section 7 concludes with a summary of this paper.

## 2. Related Work

Multi-UAV task scheduling is a typical combinatorial optimization problem. It constructs the objective function in accordance with the task requirements and UAV performance and reasonably allocates the task sequence for each UAV under the constraints of mileage and energy consumption to maximize the overall efficiency of the multi-UAV system. In the face of complex task environments and various task scheduling models, designing effective solution algorithms is the key to enhancing the task execution capabilities of multiple UAVs. In the existing research work, the methods based on optimization [18], heuristics [19], and swarm intelligence [20] are all effective solutions to this kind of problem.

The optimization-based task scheduling method is to calculate all possible solutions satisfying the constraints of UAVs and tasks in the finite solution space and to find the optimal solution as the final assignment result, that is, to find the exact solution in the finite space. Zhang et al. [21] modeled a path planning problem as a nonlinear optimal control problem with non-convex constraints and proposed a solving algorithm by approximating the non-convex parts. Yao et al. [18] formulated a joint optimization problem of the task allocation and the flying control as a mixed integer non-linear programming (MINLP) problem and minimized the drone's journey time constrained by the battery capacity and task deadlines. In order to achieve optimal task allocation with the differential-and-distortion geo-obfuscation, Wang et al. [22] built a mixed-integer non-linear model to minimize the expected travel distance of the selected workers. Lippi et al. [23] proposed a mixed-integer linear programming model to address a task allocation problem for human multi-robot settings. This model could minimize the overall execution time while optimizing human and robotic workload. You et al. [24] formulated an optimization model with the aim of minimizing the energy consumption of all UAVs under the constraints of task deadline and computing resources and proposed an iterative algorithm by applying block coordinate descent methods to solve it. For the task allocation problem of heterogeneous UAVs, Chen et al. [25] established a multi-constraint linear programming model and proposed an adaptive clustering-based algorithm to obtain approximate optimal point-to-point paths for UAVs. On this basis, they also designed a symbiotic organisms search-based optimization strategy [26] to plan the execution sequence to minimize the time consumption of the tasks.

Heuristic-based algorithms are another kind of effective solution for task scheduling problems with multiple objectives and constraints, which can find satisfactory solutions in polynomial time. Aiming at the issue of UAV coordinated scheduling, Wu et al. [27] designed a hybrid simulated annealing algorithm to obtain a scheduling scheme that took

into account the interests of both task requesters and task assigners. Zhu et al. [28] developed an efficient hybrid particle swarm optimization with a simulated annealing algorithm, which produced a high-quality solution for the rapid-assessment task-assignment problem. For Ant Colony Optimization (ACO), Wu et al. [19] proposed a dynamic labor division model to solve the UAV swarm task allocation problem by designing the dynamic task stimulus and response threshold. In addition, Zhen et al. [29] put forward an improved distributed ACO algorithm for the search–attack task scheduling for a multi-UAV platform, which showed robustness through numerical experiments. Focusing on the path planning problem of heterogeneous UAVs, Chen et al. [30] proposed an ACO-based algorithm to seek approximately optimal solutions and minimize the time consumption of tasks in the cooperative search system. For the multi-UAV task scheduling, Liu et al. [31] proposed a tabu-list-based simulated annealing algorithm to realize task allocation among multiple UAVs. They then adopted the variable neighborhood descent algorithm to generate a satisfactory scheduling scheme. Huang et al. [32] proposed an iterated heuristic framework to periodically schedule tasks. They employed the Roulette-based flight dispatching approach and a simulated annealing-based local search method to optimize the solutions.

In the application of swarm intelligence algorithms, Chen et al. [20] modified the two-part wolf pack search algorithm, which was applied to the time-sensitive multi-UAV cooperative task allocation problem. Inspired by the collaborative hunting behavior of a wolf pack, Hu et al. [33] proposed a distributed self-organizing method for UAV swarm search–attack mission planning. This method could realize flexible motion planning and group task coordinating. For the cooperative surveillance task, Liu et al. [34] established a cooperative model based on moving cost and formation stability and used a sparrow search algorithm (SSA) with fast convergence speed and strong optimization capability to solve it. Aiming at the multi-UAV search task, Fei et al. [35] devised a cooperative architecture oriented to local communication networks and proposed an improved SSA to enhance the formation optimization capability. Duan et al. [36] proposed a dynamic discrete pigeon-inspired optimization algorithm to handle cooperative search–attack missions, which could balance between benefit and consumption under the validity of constraints. Zhou et al. [37] designed an intelligent UAV swarm-based cooperative algorithm for consecutive target tracking and physical collision avoidance. Yu et al. [38] developed an improved genetic algorithm to solve the cooperative mission planning problem, which designed an efficient logic-based unlocking mechanism for the crossover and mutation operations.

The majority of the methods noted above can be used to address static scheduling issues. Table 1 provides a list of several representative methods to more effectively illustrate their attributes. These methods are frequently applied in the task pre-scheduling phase of multiple UAVs. However, the task pre-scheduling scheme will be affected by the dynamic changes of the environment and the limitations of UAV capabilities during the process of task execution, so this scheme needs to be dynamically adjusted in accordance with the environment. Thus, aiming at the above issues, we develop a method of rolling-inspired scheduling for emergency tasks by heterogeneous UAVs (RISE). This method can enhance the capabilities of rapid response and task execution for multiple UAVs in a dynamic environment.

**Table 1.** The basic attributes of the representative methods.

| Method | Literature | Tasks | UAVs | Dynamic Tasks | Base Station |
|---|---|---|---|---|---|
| Optimization | [18] | Single type | Heterogeneous | No | No |
| | [24] | Single type | Heterogeneous | Yes | No |
| | [25] | Single type | Heterogeneous | No | No |
| Heuristics | [30] | Single type | Heterogeneous | No | No |
| | [31] | Single type | Heterogeneous | No | No |
| | [32] | Single type | Heterogeneous | Yes | Yes |
| Swarm intelligence | [34] | Single type | Heterogeneous | No | No |
| | [36] | Multi-type | Heterogeneous | No | No |
| | [38] | Multi-type | Heterogeneous | No | Yes |

## 3. System Framework and Problem Statement

In this section, the framework of RISE and the primary notations utilized in this work are briefly summarized. In addition, the main issues with emergency task scheduling are briefly discussed, as well as the basic task model and UAV model. Before describing the framework and method, we first introduce the notation in this work. The main notations are summarized in Table 2.

**Table 2.** The definition of notations.

| Notation | Definition |
|---|---|
| $T_i, T_j, T_k$ | the $i$th task, $j$th task, and $k$th task |
| $U_i$ | the $i$th UAV |
| $B_i, B_q$ | the $i$th base station and $q$th base station |
| $PT_i, RT_i, at_i, dl_i$ | the task position, number of resource requirements, arrival time, and deadline of task $T_i$ |
| $PU_i, RU_i, MM_i, VU_i, ft_i$ | the UAV position, number of resources, maximum flight mileage, UAV velocity, and flight time of $U_i$ |
| $st_i$ | the service time of $B_i$ |
| $v_k$ | the initial value of $T_k$ |
| $pr_k$ | the completion probability of $T_k$ |
| $\mu_i$ | the energy consumption of $U_i$ per unit flight distance |
| $d_{i,j,k}$ | the Euclidean distance from $T_j$ to $T_k$ for $U_i$ |
| $d_{k,q}^{\min}$ | the shortest plane distance from $T_k$ to $B_q$ |
| $D_c$ | the cutoff distance |
| $LF_{j,k}^{norm}, type_{j,k}^{norm}$ | the life cycle difference and type difference between $T_j$ and $T_k$ after regularization |
| $dis_{j,k}^{norm}$ | the distance between $T_j$ and $T_k$ after regularization |
| $LF_{\max}, LF_{\min}$ | the maximum difference and minimum difference between the life cycles of $T_j$ and $T_k$ |
| $dis_{\max}, dis_{\min}$ | the maximum distance and minimum distance between $T_j$ and $T_k$ |
| $CV_i$ | the capability value of $U_i$ |
| $RV_i$ | the requirement value of $i$th task sequence |
| $L_i$ | the length of $i$th task sequence |

### 3.1. Framework

The emergency task scheduling of multiple UAVs is a complicated optimization issue with various constraints in consideration of task attributes and UAV capability. Figure 1 depicts the RISE framework in its entirety. For a given task set, an emergency scheduling process is designed via the following two phases: the task clustering phase and the optimization scheduling phase.

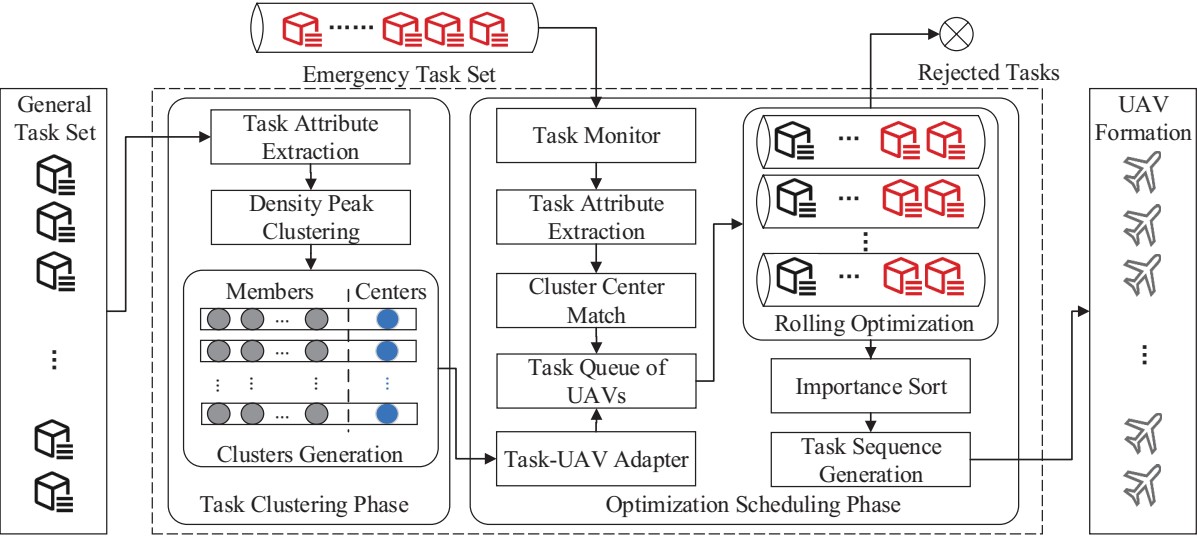

**Figure 1.** The framework of RISE.

The first phase of the RISE framework determines the classification of existing tasks, taking into account multiple inherent attributes of tasks (e.g., task position, task resource requirements, and deadlines, etc.). A DP-based algorithm is used to obtain the clustering results of the tasks in a task set. These tasks are divided into different clusters, each with its own cluster center.

The second phase of the framework delineates the allocation and scheduling of tasks-to-UAV. The tasks in each cluster are placed in a rolling queue, waiting for the UAV to complete them. The task monitor continuously monitors the system. The emergency tasks are divided into different clusters based on their attributes when they arrive at the system. These tasks are sorted according to their importance scores before being added to the rolling queues with the unscheduled tasks. Finally, the task scheduler assigns tasks in the rolling queues to different UAVs. The task will be rejected if it fails to be completed within its deadline.

### 3.2. Problem Statement

A fundamental problem of multiple UAV task scheduling is determining which UAVs are assigned to which tasks. In addition, a number of limits should be considered, including the service time of the tasks and the UAV energy consumption and flight distance, among others. Especially when emergency tasks arrive, due to the timeliness requirement, it is critical for the system to produce new scheduling schemes as rapidly as possible to ensure that these tasks are completed within their deadline. This section mainly discusses the assumptions based on the aforementioned problems, as well as the basic model in the scheduling process. We are dedicated to developing a scheduling model for multi-UAV rescue tasks in this study. In contrast to general scenarios, it is important to take into account the issue of UAVs carrying rescue materials when designing the model for rescue tasks [39,40]. As each task point has different requirements for materials and the load capacity of each UAV is limited, these two issues need to be reflected in the model.

#### 3.2.1. Problem Assumption

The formulation of the task scheduling problem is based on the following assumptions:

(1) Each task can be accomplished by at most one UAV.
(2) The assignment result of UAVs is regarded as the access to the positions of a series of task points.
(3) When the UAVs land at the base stations, they will finish replenishing the resources within a fixed time period (only considering two types of resources: battery and supply).

(4) UAVs carry out tasks at the same altitude. The 2D coordinates of rescue tasks are considered to improve the timeliness of task allocation.

### 3.2.2. Basic Model

There are three basic models in the system: task model, UAV model, and base station model. These models define the essential components of the system and quantify their various properties. In our system, we feature every model by specifying the following attributes.

(1) Task model: there are two sorts of scheduled tasks in the system: general tasks and emergency tasks. The urgency of emergency tasks is greater and their deadlines are shorter than general tasks: $GT = \{T_1, T_2, \ldots, T_{M'}\}$ indicates the general task set, and $ET = \{T_{M'+1}, T_{M'+2}, \ldots, T_M\}$ indicates the emergency task set. Both types of tasks have five characteristics. Each can be modeled by $T_i = < ID_i^T, PT_i, RT_i, at_i, dl_i >$ where they are represented in order by task number, task position, the number of resource requirements, arrival time, and deadline of task $T_i$, respectively.

(2) UAV model: the platform model in the system is defined as a group of heterogeneous UAVs $UAV = \{U_1, U_2, \ldots, U_N\}$. To further describe the characteristics of the UAV formation, each UAV can be featured by a tuple $U_i = < ID_i^U, type_i^U, PU_i, RU_i, MM_i, VU_i, ft_i >$ in which they indicate UAV number, UAV type, UAV position, the number of resources, maximum flight mileage, UAV velocity, and flight time, respectively.

(3) Base station: the role of the base station is to provide resources for UAVs, mainly including batteries and rescue materials. Let $BS = \{B_1, B_2, \ldots, B_Q\}$ be the set of base station, each station can be represented by $B_i = < ID_i^B, PB_i, st_i >$, where $ID_i^B$ is the station number, $PB_i$ is the station position, and $st_i$ is the service time of $B_i$.

We also define a decision variable $\eta_{i,j,k}$, which is a binary variable, to determine whether $T_j$ and $T_k$ can be carried out by $UAV_i$; $\eta_{i,j,k}$ can be expressed as:

$$\eta_{i,j,k} = \begin{cases} 1, & \text{if the } UAV_i \text{ can perform from } T_j \text{ and } T_k, \text{ and } j \neq k \\ 0, & \text{otherwise} \end{cases} . \tag{1}$$

## 4. Modeling and Formulation

Because the emergency task scheduling of multiple UAVs is a NP-hard problem, we establish an optimization model to solve it. The proposed model is composed of two parts: objective function and constraints. This section focuses on the definition process of the objective function as well as a thorough analysis and formulation of the model constraints.

### 4.1. Objective Function

It is vital to ensure not only that as many tasks as possible can be accomplished during the task execution of UAV formation, but also that the formation execution efficiency is maintained. Thus, to solve the scheduling problem of emergency tasks with dynamic arrival and short life cycle, we design an objective function that includes task profit, flight energy consumption, and safety risk by thoroughly considering the task profit and UAV flight cost in the process of task execution. Next, we calculate the best task sequence for each UAV to ensure the formation's task execution efficiency. The objective function of RISE is described in detail below.

**Definition 1.** *Task Profit. Each task is given an initial value based on its importance, and the number of resource requirements is used to predict the probability of task completion. Thus, task profit is defined as follows:*

$$\text{PF} = \sum_{i=1}^{N} \sum_{j=0}^{M} \sum_{k=1}^{M} \eta_{i,j,k} \times v_k \times pr_k, \tag{2}$$

*where $v_k$ represents the value of $T_k$, which reflects the task's importance, the value of $v_k$ is a random number between $[0, 1]$: the smaller the $v_k$, the lower the task's value; $pr_k$ indicates the probability*

*that $T_k$ is successfully executed, $pr_k = 1 - RU_k/(\max\{RU_l | l \in [1, M]\} + 1)$, which indicates that the more resource requirements of the task, the lower $pr_k$ of the task. The higher the value of PF, the more rational the order of UAV task execution is, and the probability of task completion rises accordingly.*

**Definition 2.** *Fight Energy Consumption. Low flight cost is one of the crucial goals for the UAV formation due to the capacity limitation of airborne batteries, which impacts task execution efficiency. Flight energy consumption refers to the consumption generated in the process of the UAV from the starting position to the task position, including battery energy consumption, equipment loss, and other factors. It is defined as follows:*

$$\text{EC} = \sum_{i=1}^{N} \sum_{j=0}^{M} \sum_{k=1}^{M} \eta_{i,j,k} \times \mu_i \times d_{i,j,k}, \tag{3}$$

*where $\mu_i$ is the energy consumption of $UAV_i$ per unit flight distance, and $d_{i,j,k}$ represents the Euclidean distance from $T_j$ to $T_k$ for $UAV_i$. The lower EC, the lower the flight cost of the UAV formation and the higher the task execution efficiency.*

**Definition 3.** *Safety Risk. Flight safety is a factor that must be considered when UAVs perform missions, and it needs to be analyzed from the level of the whole flight route. All UAVs take off from the same initial position and perform the assigned tasks, respectively. Each UAV is independent of the others according to Assumption 1. The safety risk for any UAV refers to the flight risk assessment value affected by the length of the task queue and the distance between adjacent tasks. It is defined as follows:*

$$\text{FR} = \sum_{i=1}^{N} \sum_{j=0}^{M} \sum_{k=1}^{M} \eta_{i,j,k} \times \frac{d_{i,j,k}}{MM_i}. \tag{4}$$

It can be seen from (4) that the greater the number of tasks and the longer the flight distance, the greater FR and the lower the flight reliability of the UAV formation. Consequently, based on the three factors above, the objective function of RISE is:

$$\max \text{F}_{RISE} = \max(\alpha_1 \text{PF} - \alpha_2 \text{EC} - \alpha_3 \text{FR}), \tag{5}$$

where $\alpha_1$, $\alpha_2$, and $\alpha_3$ are normalization coefficients.

*4.2. Constraints*

Multiple UAVs are limited by various constraints during task execution. These constraints and their formulation are described in detail in this section, involving task time constraints, UAV flight constraints, and energy consumption constraints. The specific descriptions are as follows:

(C1) Each UAV starts from the initial position to perform the assigned tasks. If tasks are accepted, the UAV will take off from the initial position and fly to the first task position. The UAV will stay at the initial position for task preparation when it does not receive the tasks. Constraint C1 can be formulated as follows:

$$\sum_{j=1}^{M} \eta_{i,0,j} \leq 1, \forall i \in [1, N], \tag{6}$$

where $\eta_{i,0,j}$ indicates $UAV_i$ takes off from the initial position and carries out the $j$th task.

(C2) Each task can be carried out by at most one UAV. The purpose of C2 is to avoid repeated task execution, which will affect the overall efficiency of the UAV formation.

For $T_j$, the number of arriving and departing UAVs is less than or equal to one. Constraint C2 is expressed as:

$$\sum_{i=1}^{N}\sum_{j=0}^{M}\eta_{i,j,k} = \sum_{i=1}^{N}\sum_{l=1,l\neq k}^{M}\eta_{i,k,l} \leq 1, \forall k \in [1, M]. \tag{7}$$

Specially, if $UAV_i$ does not carry out $T_k$, the number of times $UAV_i$ arrives and leaves $T_k$ is $\sum_{i=1}^{N}\sum_{j=0}^{M}\eta_{i,j,k} = \sum_{i=1}^{N}\sum_{l=1,l\neq k}^{M}\eta_{i,k,l} = 0$. If $UAV_i$ carries out $T_k$, the number of times $UAV_i$ arrives and leaves $T_k$ is equal to 1 ($\sum_{i=1}^{N}\sum_{j=0}^{M}\eta_{i,j,k} = \sum_{i=1}^{N}\sum_{l=1,l\neq k}^{M}\eta_{i,k,l} = 1$).

(C3) The system requires that each UAV only perform one task at a time to ensure that it can carry out tasks reliably and effectively, that is, the task sequence assigned to the UAV is executed in order. We use a two-dimensional matrix $S = \{s_{i,j}|1 \leq i \leq N, 0 \leq j \leq M\}$ to record the execution order of each UAV. The expression $s_{i,k}$ is the integer variable, which can be expressed as:

$$s_{i,k} = \begin{cases} s_{i,j} + 1, & \text{if } \eta_{i,j,k} = 1 \\ 0, & \text{if } \eta_{i,j,k} = 0 \end{cases}, \quad \begin{array}{l} \forall i \in [1, N], \forall j \in [0, M] \\ \forall k \in [1, M], k \neq j \end{array}, \tag{8}$$

where $s_{i,0} = 0$ represents $UAV_i$ taking off from the initial position. For the task execution order of $UAV_i$, $s_{i,k}$ should be less than or equal to the total number of tasks M. The expression is as follows:

$$s_{i,k} \leq M, \forall i \in [1, N], \forall k \in [1, M]. \tag{9}$$

(C4) Each UAV must perform tasks within a given mileage range. The UAV takes off from the initial position to execute tasks, and energy is continuously consumed with the increase in flight mileage. As the battery carried by the UAV is fixed, the UAV needs to be refueled with energy when confronted with the dynamic arrival tasks. The base stations are set up in the task region for the UAV's energy provisioning. Assume that $UAV_i$ departs from the position of $T_j$, it must determine whether the current remaining mileage is sufficient to reach the position of $T_k$, and C4 is formulated as follows:

$$\begin{cases} \sum_{j=0}^{M}\eta_{i,j,k}(d_{i,j,k} + d_{k,q}^{\min}) \leq MM_i \\ d_{i,j,k} = \text{sqrt}((x_j^T - x_k^T)^2 + (y_j^T - y_k^T)^2) \\ d_{k,q}^{\min} = \min(\text{sqrt}((x_k^T - x_q^B)^2 + (y_k^T - y_q^B)^2)) \end{cases} \quad \begin{array}{l} \forall i \in [1, N], \forall j \in [0, M] \\ \forall k \in [1, M], \forall q \in [1, Q] \end{array}, \tag{10}$$

where $d_{k,q}^{\min}$ is the shortest flight distance from $T_k$ to $B_q$, and $d_{i,j,k}$ is the distance from $T_j$ to $T_k$ for $UAV_i$.

(C5) Only if the UAV meets the premise that its number of resources is greater than the demand for task resources can it carry out the task. Due to the different types of tasks, the consumption of UAV resources is also different. As the total resources of the UAV are constantly consumed, $UAV_i$ needs to determine whether the resources it carries meet the requirements of task execution in advance before executing $T_k$. If this condition is not met, $UAV_i$ needs to fly to the base station to supplement resources. Constraint C5 can be formulated as:

$$\sum_{j=0}^{M}\eta_{i,j,k}RT_k \leq RU_i, \forall i \in [1, N], \forall k \in [1, M]. \tag{11}$$

(C6) Each task is required to be completed within its deadline. The dynamically reached task has stringent time constraints, and the UAV must complete it within the

specified time. The task will be rejected if it is not completed within the time limit. The time it takes for $UAV_i$ to fly from $T_j$ to $T_k$ is $d_{i,j,k}/VU_i$; C6 can be formulated as:

$$TU_i + \sum_{j=0}^{M} \eta_{i,j,k} \times d_{i,j,k}/VU_i \leq dl_k, \forall i \in [1,N], \forall k \in [1,M], \tag{12}$$

where $TU_i$ represents the total flight time of the completed tasks for $UAV_i$.

Based on the above definitions and descriptions of objective functions and constraints, we conduct an optimization model of task scheduling, which is formulated as follows:

$$\begin{aligned} \max \mathrm{F}_{RISE} &= \max(\alpha_1\mathrm{PF} - \alpha_2\mathrm{EC} - \alpha_3\mathrm{FR}) \\ \text{s.t.} \quad &C1, C2, C3, C4, C5, C6 \end{aligned}. \tag{13}$$

Although the above linear programming problem can obtain the exact solution, the complexity and difficulty of addressing this problem increases as a large number of tasks arrive at the system dynamically, especially if there are certain emergency tasks, and the timeliness of this problem cannot be guaranteed. On the basis of the above facts, we propose the optimization algorithm RISE.

## 5. Emergency Task Scheduling Method

We devise RISE to generate a high-quality scheduling scheme that includes two phases: task clustering and optimization scheduling. Task clustering is critical to the whole task-to-UAV scheduling process. The reason for this is that the task scheduling problem is a multi-constraint optimization problem with exponentially increasing solution space and time complexity as the number of tasks increases. Moreover, task attributes such as geographical position, resource requirements, and deadline have an impact on UAV allocation. Hence, we use the improved DP algorithm to achieve task clustering, which improves the efficiency of UAV task allocation. In addition, we design a rolling-inspired optimization scheduling approach, which transforms the dynamic optimization problem into multiple local optimization problems. The attribute information of tasks and UAVs is fully considered to achieve the optimal scheduling of task-to-UAV.

### 5.1. Improved DP-Based Task Clustering

The density peaks clustering (DP) algorithm [41], which was published in Science, can automatically find cluster centers and achieve efficient clustering of arbitrary shaped data. It can be observed that the core of it is the design of cluster centers. The authors considered that cluster centers have two characteristics: (i) cluster centers are surrounded by neighbors with lower local density $\rho_j$; (ii) they are at a relatively large distance $\delta_i$ from any points with a higher local density. The formulations are given as:

$$\begin{aligned} \rho_j &= \sum_{k=1,k\neq j}^{M} \chi(D_{j,k} - D_c) \\ \chi(D_{j,k} - D_c) &= \begin{cases} 1, D_{j,k} < D_c; \\ 0, \text{otherwise} \end{cases}, \end{aligned} \tag{14}$$

$$\begin{aligned} \delta_j &= \begin{cases} \min_{j \in I_S^j}\{D_{j,k}\}, I_S^j \neq 0; \\ \max_{j \in [1,M]}\{D_{j,k}\}, \text{otherwise} \end{cases}, \\ I_S^j &= \{k \in [1,M] : \rho_k > \rho_j\} \end{aligned} \tag{15}$$

where $D_c$ is the cutoff distance, which can be selected so that the average number of neighbors is 2% of the total number of tasks; $\chi()$ is the logical judgment function; $D_{j,k}$ is the distance between $T_j$ and $T_k$; $I_S^j$ is index set, indicating that the local density $\rho_j < \rho_k$.

The "distance" in DP does not only refer to Euclidean distance but also a broad concept, which can be composed of task's deadline and distance, etc. In our approach, we try to improve DP and build a two-dimensional distance matrix $D = \left\{ D_{j,k} | j,k \in [1.M] \right\}$ among tasks using two attributes: life cycle and position. A task's life cycle refers to a period from the time it arrives at the system to its deadline; $D_{j,k}$ is formulated as follows:

$$D_{j,k} = \lambda_1 LF_{j,k}^{norm} + \lambda_2 dis_{j,k}^{norm}, \tag{16}$$

where $LF_{j,k}^{norm}$ is the life cycle difference between $T_j$ and $T_k$ after normalization, respectively; and $dis_{j,k}^{norm}$ is the distance between $T_j$ and $T_k$ after normalization. They are expressed as follows:

$$\begin{cases} LF_{j,k} = \left| (dl_j - at_j) - (dl_k - at_k) \right| \\ LF_{j,k}^{norm} = (LF_{j,k} - LF_{min}) / (LF_{max} - LF_{min}) \end{cases}, \tag{17}$$

$$\begin{cases} dis_{j,k} = \sqrt{(x_j^T - x_k^T)^2 + (y_j^T - y_k^T)^2} \\ dis_{j,k}^{norm} = (dis_{j,k} - dis_{min}) / (dis_{max} - dis_{min}) \end{cases}, \tag{18}$$

where $LF_{max}$ and $LF_{min}$ are the maximum difference and minimum difference between the life cycles of $T_j$ and $T_k$. Similarly, $dis_{max}$ and $dis_{min}$ are the maximum distance and minimum distance between $T_j$ and $T_k$, respectively.

We use the decision factor $\gamma_j$ of DP to automatically select the cluster centers of the current task set; $\gamma_j$ requires that each center have a high local density and a greater distance from other high-density tasks. It can be defined as follows:

$$\gamma_j = \rho_j \delta_j, \forall j \in [1, M]. \tag{19}$$

The number of cluster centers depends on the number of UAVs in the system. We select the same number of cluster centers and UAVs, so that each UAV can perform a group of tasks with similar attributes; $\{\gamma_j\}_{j=1}^{M'}$ of all tasks are sorted in descending order, and the top $N$ tasks with the highest $\gamma$ are chosen as the cluster centers of the current task set.

It is critical to classify other task members after obtaining the cluster centers. If the nearest neighbor strategy of DP is used to partition these members, the number of them in each cluster will be greatly different. A large number of members will easily cause the UAV to take too long to perform such tasks and be unable to handle the emergency tasks that arrive to the system in a timely manner, resulting in a reduction in the task completion rate. Conversely, a small number of members might quickly lead to the UAV becoming idle when executing such tasks, and the UAV can rapidly complete these tasks and remain idle in hover until new tasks are assigned, resulting in lower platform utilization. As a result, we propose a balanced allocation mechanism to improve DP in the process of partitioning members, so as to ensure that the number of members in each cluster is balanced as much as possible. For the current task set $T$, the average value of the number of tasks to be assigned to each UAV is formulated as follows:

$$AVG = \lceil M/N \rceil. \tag{20}$$

In the process of nearest neighbor partition, it is required to limit the boundary density of each cluster, i.e., the number of cluster members, during the nearest neighbor partition procedure. For each cluster center $c_i, i \in [1, N]$, $T_k$ is divided into corresponding cluster if $D_{c_i,k}$ satisfies:

$$D_{c_i,k} \leq D_{c_i,AVG}, \forall k \in [1, M]. \tag{21}$$

The improved DP algorithm has the following characteristics:

(1) It can effectively avoid the generation of halo points and ensure the accuracy and rationality of task clustering.

(2)    The number of members in each cluster is more balanced, which can effectively solve the problems of low platform utilization and low task completion rate.

Figure 2 depicts task clustering based on improved DP with fifteen tasks as examples. The tasks are located on a two-dimensional map, which is shown in Figure 2a, and their positions are randomly generated. Next, $\rho_j$ and $\delta_i$ are calculated using improved DP to generate a decision graph. We employ (20) to evaluate the quality of cluster centers. From Figure 2b, we can see that $T_4$, $T_5$, $T_6$, and $T_{15}$ are selected as cluster centers. Once the centers are determined, we utilize (20) and (21) to partition the remaining tasks to ensure that the number of members in each cluster is as balanced as possible. The results are shown in Figure 2c.

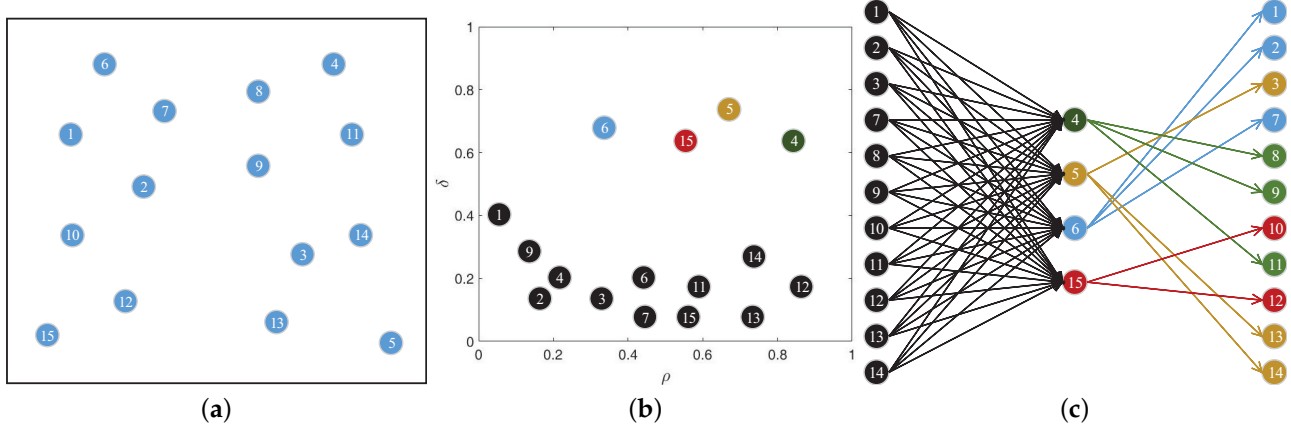

(a)                                      (b)                                      (c)

**Figure 2.** Illustrations of task clustering phase: (**a**) task layout; (**b**) center selection; (**c**) task partition.

### 5.2. Rolling-Inspired Optimization Scheduling

Three key parts in the optimization scheduling phase are devised to enable RISE to achieve efficient task scheduling, namely, task-UAV adapter, emergency task manager, and rolling queue generator. The adapter is responsible for completing capability mapping between UAV and a cluster of tasks, which means providing a UAV with a task queue after clustering. The manager monitors the emergency tasks arriving at the system in a real-time manner and assigns them to the appropriate task cluster. The most significant part of this phase is the rolling queue generator, which can determine the execution order of tasks in the queue and dynamically adjust the order as required. The detailed descriptions are as follows.

#### 5.2.1. Task-UAV Adapter

The requirement value of the task sequence and the capability value of the UAV should be evaluated to ensure that the UAV formation can effectively carry out these tasks. To meet the requirements of task scheduling for timeliness and completion rate, the crucial attributes ($RU_i$ and $MM_i$) are used to assess the capability value of the UAV. It can be defined as follows:

$$CV_i = \frac{MM_i}{\sum\limits_{j=1}^{N} MM_j} + \frac{RU_i}{\sum\limits_{j=1}^{N} RU_j}, \forall i \in [1, N], \tag{22}$$

where $CV_i$ indicates the capability value of $U_i$; the larger $CV_i$ implies that $U_i$ has superior task execution capability. Due to the heterogeneous nature of UAVs, we arrange all UAVs in descending order according to their capability values to get a list of UAVs. The task queues are assigned to UAVs based on this list, which can dramatically enhance the execution

efficiency of UAVs. Similarly, the requirement value of the task sequence $RV_i$ is formulated as follows:

$$RV_i = \left[ \sum_{l=1}^{L_i} (dl_l - at_l) / \sum_{k=1}^{M} (dl_k - at_k) + RT_l / \sum_{k=1}^{M} (RT_k) \right] / L_i, \forall i \in [1, N], \quad (23)$$

where $L_i$ is the length of $i$th task sequence; $dl_l - at_l$ represents the life cycle of $T_l$: it can reflect the task urgency; the larger $dl_l - at_l$ is, the lower urgency is. We arrange the requirement values of all tasks in descending order and adapt them to the UAV list. The adaptation principle is that the task sequence with the largest requirement value is matched to the UAV number with the largest capability value, so as to improve the success rate of task execution.

### 5.2.2. Emergency Task Manager

Due to the shorter deadline of emergency tasks compared to general tasks, UAVs should be arranged to execute such tasks as soon as they arrive at the system. The task surveillance module in the emergency task manager is set to detect the arrival of the emergency tasks in a real-time manner. After receiving the emergency tasks, the attribute $PT$ of these tasks are extracted and quickly matched with the cluster centers $Cluster = \{Clu_1, Clu_2, \ldots, Clu_N\}$ obtained by the improved DP algorithm, so as to generate the task sequence for each UAV at the current time. The match score $MS_{i,j}$ can be formulated as follows:

$$MS_{i,j} = \min \frac{\left(x_i^{Clu} - x_j^{ET}\right)^2 + \left(y_i^{Clu} - y_j^{ET}\right)^2}{\sum_{i=1}^{N} \left(x_i^{Clu} - x_j^{ET}\right)^2 + \left(y_i^{Clu} - y_j^{ET}\right)^2}, \forall j \in [1, M - M'], \quad (24)$$

where $M - M'$ is the number of the emergency tasks. For each emergency task, $MS_{i,j}$ is calculated based on task position, which is inversely proportional to the distance between the task and the cluster center. The emergency task $T_j$ is quickly assigned to the sequence of the cluster center with the smallest distance from it.

### 5.2.3. Rolling Queue Generator

Each UAV is assigned a task sequence after the previous two parts are completed. Many tasks may be rejected due to exceeding their life cycle if only the tasks in the sequence are executed in order, resulting in a low completion rate of tasks. Aiming at this issue, the rolling queue generator is designed to optimize the task sequence of each UAV. The execution order of tasks will be modified according to their attributes such as urgency and location, so as to improve the completion rate of these tasks.

We arrange the tasks in the sequence in ascending order according to the deadline during the process of rolling queue generation, so that the task with the highest urgency can be completed first, meeting constraint C6. If C6 is not satisfied, this task will be assigned to other UAVs that meet C6. Furthermore, this task will be rejected if no UAV in the formation can meet its requirements.

In addition, there is still another issue to be resolved. When facing scheduled tasks with the same deadline, it is vital to comprehensively analyze the profits and costs of executing tasks so as to optimize the execution order of these tasks. However, due to the limitation of a UAV's maximum mileage and resource capacity, we devise the rolling queue in the optimization scheduling and arrange tasks with the same deadline in descending order of importance (max $F_{RISE}$), so as to generate the final task execution order. Suppose that there are $m$ tasks $T^{RL} = \{T_1^{RL}, T_2^{RL}, \ldots, T_m^{RL}\}$ in the rolling queue at this moment for $UAV_i$.

(1) If $T_1^{RL}$ fails to meet C4, $UAV_i$ cannot perform tasks. This phenomenon indicates that $UAV_i$ needs to fly to the nearest base station for supplemental energy. Because its position has changed after arriving at the base station, the tasks in the rolling queue must be reordered so that the task with the highest importance score is completed first.

(2) If $T_1^{RL}$ fails to meet C5, $UAV_i$ cannot perform tasks. There are two factors that contribute to this occurrence. One is that if the initial resource of $UAV_i$ is insufficient to meet the requirement of $T_1^{RL}$, then $T_1^{RL}$ should be assigned to other UAVs that meet C5 and C6, and it will be deleted from the queue. If not, $T_1^{RL}$ will be rejected. The second is that the $UAV_i$ needs to fly to the nearest base station for supplemental energy. The tasks in the rolling queue need to be reordered just as in (1).

(3) If $T_1^{RL}$ meets C4 and C5, $UAV_i$ is scheduled to carry out this task, and $T_1^{RL}$ will be removed from the queue.

The following are the advantages of the rolling queue generator:

(1) The dynamic optimization problem is transformed into multiple local optimization problems by fully considering the profits and costs of task execution, which can effectively reduce the cost of optimal scheduling and enhance task execution feasibility.

(2) The tasks with the earliest deadline are scheduled first to ensure the overall success rate of task execution.

(3) Tasks are awaiting execution in the rolling queue. There is no migration overhead caused after rescheduling, effectively lowering the energy consumption of system scheduling.

The optimization scheduling process is shown in Figure 3. When the emergency tasks $T_6$, $T_7$, and $T_8$ arrive in the system, they are quickly assigned to the UAV that meets their requirements according to the task-UAV adapter. They are inserted into the task queue waiting for scheduling according to their deadlines. After that, the rolling queue is sorted according to the importance score of each task and generates a new task scheduling scheme; the positions of $T_6$, $T_7$, and $T_8$ in the queue are readjusted at this moment. In the rolling queue, $T_4$ and $T_5$ are assigned to other UAVs if they could not be completed before their deadlines. If all UAVs are unable to perform these two tasks, they will eventually be rejected by the UAV formation.

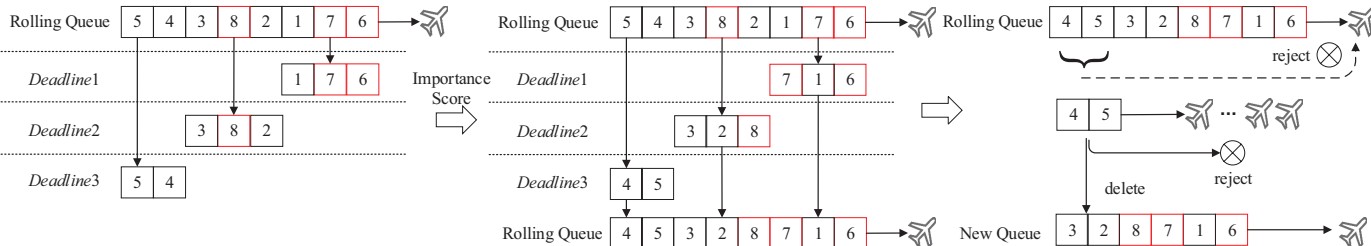

**Figure 3.** Illustration of task clustering phase.

### 5.3. Algorithm Implementation

The main framework of the proposed RISE is described in Algorithm 1. First, the essential parameters, which include attributes of tasks, UAVs, and base stations, are initialized. The improved DP algorithm is used to realize the matching between the UAV and a cluster of tasks (steps 2–3). Subsequently, the main loop of RISE is proposed. For each iteration, the emergency task is assigned into the task sequence, then Rolling Optimization() is enabled to achieve optimal scheduling of emergency tasks (steps 5–10).

---

**Algorithm 1:** RISE

---

1  Parameter initialization of UAVs, tasks, and base stations;
2  Improved DP($N$,$M'$,$\{T_j\}_{j=1}^{M'}$);
3  Adapt $\{T_j\}_{j=1}^{M'}$ to $\{U_i\}_{i=1}^{N}$;
4  $flag_T \leftarrow 1$;
5  **while** $flag_T$ **do**
6     **if** *Emergency task* $\{ET_j\}_{j=1}^{M-M'}$ *arrives* **then**
7          Assign the emergency task to the task sequences of UAVs by (24);
8      Rolling Optimization($N$, $\{Task_U\}_{i=1}^{N}$);
9     **if** *the task sequences of UAVs are* $\varnothing$ **then**
10         $flag_T \leftarrow 0$;

---

The detailed implementation procedure of the improved DP is as follows. As shown in Algorithm 2, the distance matrix $D$ is constructed by (16) for existing tasks after initializing parameters. The cutoff distance $d_c$ is determined according to $D$, and the local density $\rho_j$ and distance $\delta_i$ of each task are calculated by (14) and (15), respectively. To evaluate the quality of cluster centers, the decision factor $\gamma_j$ is calculated by (19), and $\{\gamma_j\}_{j=1}^{M'}$ of all tasks are sorted in descending order. After sorting, we select the top $N$ tasks with the highest $\gamma$ as the cluster centers (steps 11–18). Subsequently, the remaining tasks are divided into the closest cluster by calculating the distance from each center, and the proposed balanced allocation mechanism is used to balance the number of members in each cluster (steps 19–30).

The implementation of the rolling optimization phase is shown in Algorithm 3. The inputs of this algorithm are composed of the number of UAVs, the task sequences of UAVs, and the task set. For each UAV, the tasks in the sequence are arranged in ascending order according to their deadlines, and then the execution order of tasks with the same deadline is determined by calculating their importance. When the resources and flight distance of $U_i$ are sufficient to meet the execution of $T_\varphi$, $T_\varphi$ is arranged into $SS_i$ (steps 8–13). If the distance between $U_i$ and $T_\varphi$ is greater than the maximum flight mileage, the UAV should fly to the nearest base station, followed by the tasks in the sequence being reordered according to Importance Sort() (steps 14–17). When the resources of $U_i$ fail to satisfy the requirements of $T_\varphi$, the UAV also needs to fly to the station. However, if the UAV is still unable to complete, $T_\varphi$ will be assigned to another UAV. Finally, $T_\varphi$ is rejected if no UAV can perform it.

---

**Algorithm 2:** Improved DP()

---

1 **Input:** The number of clusters $N$; The number of tasks $M'$; Task set $\{T_j\}_{j=1}^{M'}$;

2 **Output:** Task label set $\{TL\}_{j=1}^{M'}$;

3 **for** $j = 1$ *to* $M' - 1$ **do**

4     **for** $k = j + 1$ *to* $M'$ **do**

5         Construct the distance matrix $D$ by (17);

6 Determine cutoff distance $d_c$ according to matrix $D$;

7 Calculate the local density $\{\rho_j\}_{j=1}^{M'}$ of the task by (15);

8 Calculate the distance $\{\delta_j\}_{j=1}^{M'}$ of the task by (16);

9 Calculate the decision factors $\{\gamma_j\}_{j=1}^{M'}$ by (19);

10 Sort $\{\gamma_j\}_{j=1}^{M'}$ in descending order;

11 $\{TL\}_{j=1}^{M'} \leftarrow 0$;

12 $N_{cluster} \leftarrow 1$;

13 $center = []$;

14 **for** $j = 1$ *to* $M'$ **do**

15     **if** $\rho_j \times \delta_j \geq \gamma_N$ **then**

16         $TL_j = N_{cluster}$;

17         $center = [center; N_{cluster}]$;

18         $N_{cluster} = N_{cluster} + 1$;

19 $AVG = \lceil M'/N \rceil$; $\{num\}_{i=1}^{N} \leftarrow 0$;

20 **for** $j = 1$ *to* $M'$ **do**

21     **if** $TL_j == 0$ **then**

22         $D_{TL} = max(D)$;

23         $cluster = 0$;

24         $c_{num} = 0$;

25         **for** $i = 1$ *to* $N$ **do**

26             **if** $D_{TL} \leq D(j, center(i))$ *and* $num(center(i)) \leq AVG$ **then**

27                 $D_{TL} = D(j, center(i))$;

28                 $cluster = center(i)$;

29         $TL_j = cluster$;

30         $num(cluster) = num(cluster) + 1$;

---

The pseudocode of the function Importance Sort() is given in Algorithm 4. It can be seen that the task with the minimum deadline is scheduled first (step 3), but it fails to carry out $T_j$ because the finish time of $U_i$ is less than the deadline of $T_j$. At this point, $T_j$ is assigned to $U_k$; otherwise, it will be rejected (steps 5–13). For the tasks that $U_i$ can perform, their importance scores are calculated by (5), and Algorithm 4 finally outputs the sorted results of these tasks according to importance score (steps 16–17).

---

**Algorithm 3:** Rolling Optimization()

---

1  **Input:** The number of UAVs $N$; The task sequences of UAVs $\{Task_{U_i}\}_{i=1}^{N}$;
2  **Output:** The scheduling sequences of UAVs $\{SS\}_{i=1}^{N}$;
3  **for** $i = 1$ *to* $N$ **do**
4     The task sequence of $U_i$ is arranged in ascending order according to deadline $T_{dl}$;
5     Importance Sort(task sequence of $U_i$);
6     $decision \leftarrow 1$;
7     **while** *decision* **do**
8         **if** $T_{\varphi}.RT \leq U_i.RU$ *and* $dis(T_{\varphi}, U_i) \leq U_i.MM$ **then**
9             Update $U_i.PU$ according to $T_j.PT$;
10            $U_i.MM = U_i.MM - dis(T_{\varphi}, U_i)$;
11            $U_i.RU = U_i.RU - T_{\varphi}.RT$;
12            Arrange $T_{\varphi}$ into $SS_i$;
13            $decision \leftarrow 0$;
14         **else if** $T_{\varphi}.RT \leq U_i.RU$ *and* $dis(T_{\varphi}, U_i) \geq U_i.MM$ **then**
15            $U_i$ flies to the nearest base station for supplemental energy;
16            Update $MM$, $PU$, $RU$ and $ft$ of $U_i$;
17            ImportantSort(task sequence of $U_i$);
18         **else if** $T_{\varphi}.RT \geq U_i.RU$ **then**
19            **if** $T_{\varphi}.RT \leq$ *initial* $U_i.RU$ **then**
20                $U_i$ flies to the nearest base station for supplemental energy;
21                Update $MM$, $PU$, $RU$ and $ft$ of $U_i$;
22                Importance Sort(task sequence of $U_i$);
23            **else**
24                **if** $T_{\varphi}.dl$ *and* $T_{\varphi}.RT$ *meet the requirements of* $U_k$ **then**
25                   Reassign $T_{\varphi}$ to $U_k$ and delete $T_{\varphi}$ from the task sequence of $U_i$;
26                **else**
27                   Refuse $T_{\varphi}$;
28                $decision \leftarrow 0$;

---

**Algorithm 4:** Importance Sort()

---

1  **Input:** The task sequence of $U_i$;
2  **Output:** The task number $\varphi$ with max importance score;
3  $J \leftarrow \min(T_{dl}$ of the task sequence$)$;
4  **for** $j = 1$ *to* $J$ **do**
5     **if** $T_j.dl > U_i.ft$ **then**
6         $flag = 0$;
7         **for** $k = 1$ *to* $N$ **do**
8            **if** $T_j.dl \leq U_k.ft$ *and* $k \neq i$ **then**
9                Reassign $T_j$ to $U_k$;
10               Delete $T_j$ from the task sequence of $U_i$;
11               $flag = 1$;
12         **if** $flag == 0$ **then**
13            Refuse $T_j$;
14     **else**
15         Calculate the importance score $IS_j$ of $T_j$ by (5);
16  Sort $\{IS_j\}_{j=1}^{J}$ in descending order;
17  $\varphi \leftarrow num(IS_1)$;

---

## 6. Experiment Results

In this section, we conduct a simulation experiment aimed at the rescue task scheduling problem in an open outdoor environment, which fits to the assumptions in Section 3.2.1. All our experiments are implemented on the Intel Core i7 3.00GHz PC with 16GB RAM (Lenovo, Beijing, China). The tasks in the experiment are all rescue tasks, and they are divided into general tasks and emergency tasks according to the arrival time. The initial scenario of multi-UAV task scheduling is deployed as shown in Figure 4. The size of the task region is $8 \times 12$ km, the blue points mark the positions of general tasks, and the red rectangles indicate the deployment positions of the base stations. Moreover, the required weight of each task is randomly generated in the interval (0,2] kg, and the deadline of each task is randomly selected in [300 s, 600 s, 900 s, 1200 s].

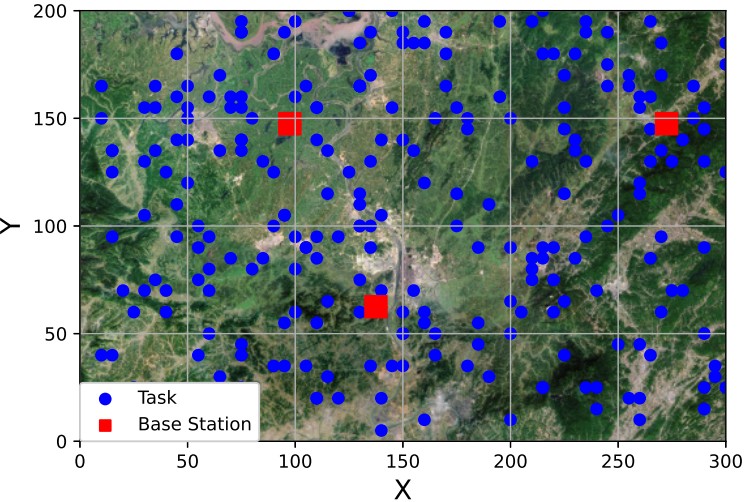

**Figure 4.** Task location deployment in an outdoor environment.

The basic information about UAVs and base stations is shown in Tables 3 and 4. It can be seen from Table 3 that six heterogeneous UAVs with different attributes are used to perform rescue tasks. The flight speed is a key factor to be considered in path planning, whereas our work is to design a reasonable task allocation scheme to achieve dynamic task scheduling. As a consequence, we set the UAV's flight speed as a fixed value in the experiment. Although they have the same initial position and flight speed, their maximum mileage and carrying resources are different. The position coordinates and service time of each base station are listed in Table 4. We assume that the process of UAVs' energy supplement is to replace the battery, so it takes a fixed time (30 s) to replenish the energy in the base stations.

**Table 3.** The basic information of the UAVs.

| ID | PU | RU(kg) | MM(km) | VU(m/s) | ft(s) |
|----|-----|--------|--------|---------|-------|
| $U_1$ | (0,0) | 25 | 100 | 50 | 0 |
| $U_2$ | (0,0) | 25 | 100 | 50 | 0 |
| $U_3$ | (0,0) | 15 | 250 | 50 | 0 |
| $U_4$ | (0,0) | 15 | 250 | 50 | 0 |
| $U_5$ | (0,0) | 20 | 200 | 50 | 0 |
| $U_6$ | (0,0) | 20 | 200 | 50 | 0 |

**Table 4.** The basic information of the base stations.

| ID | PB | st(s) |
|---|---|---|
| $B_1$ | (5.4,2.4) | 30 |
| $B_2$ | (4.0,6.0) | 30 |
| $B_3$ | (10.8,6.0) | 30 |

The main work of this paper is to address the task scheduling problem of multiple UAVs in a dynamic environment. Six UAVs begin to perform tasks after the initial task scheduling scheme is generated. Emergency tasks arrive at the system after a period of time, and these tasks are allocated to the task rolling queues of the UAVs, so the order of each group of unscheduled tasks needs to be recalculated. In our experiment, the number of general tasks is set at $M' = 200$, and the number of emergency tasks is set at $M - M' = 50$. Based on the above parameter settings, the task execution results of UAVs are shown in Figure 5. It illustrates the change in the number of unscheduled tasks for each UAV.

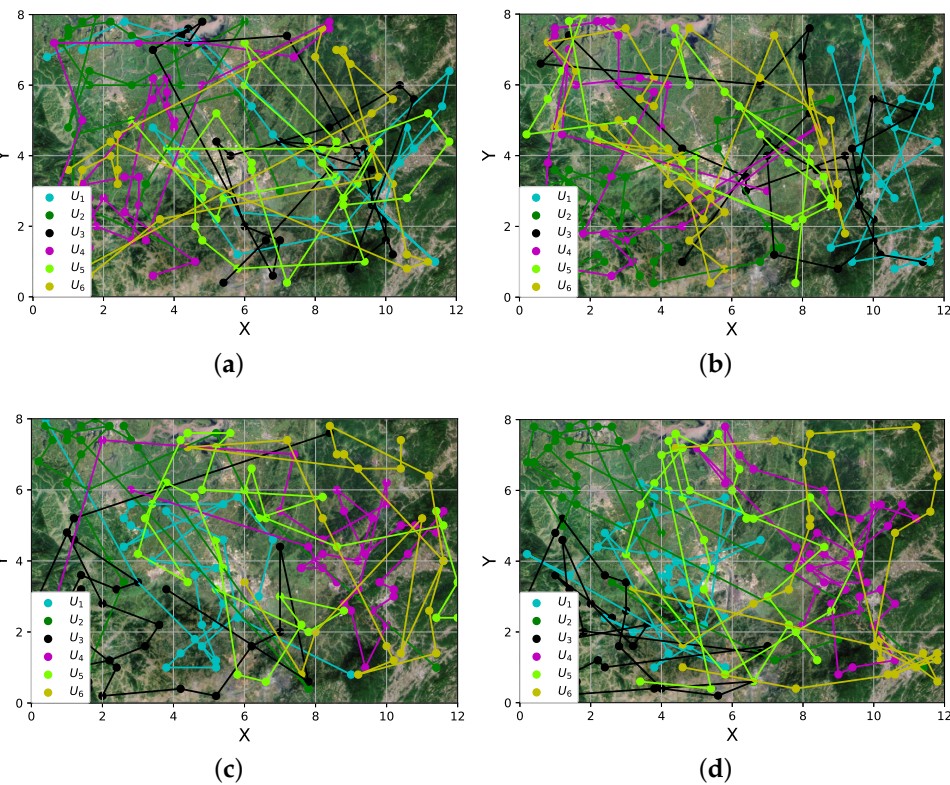

**Figure 5.** The flight routes generated by four methods: (**a**) SC; (**b**) FCM; (**c**) DPGA; (**d**) RISE.

## 6.1. Performance Evaluation

As far as we know, there is no work on dynamic scheduling of multiple UAVs for emergency tasks. We cannot compare the proposed algorithm with other complete algorithms, so we are only able to improve existing similar algorithms to fulfill the experimental requirements of our method. In this section, we select the more popular spectral clustering (SC) [42], fuzzy c-means algorithm (FCM) [43], and DPGA [25] in recent years for qualitative and quantitative analysis.

SC is a clustering method based on the concept of graph theory. It obtains clustering results by calculating the similarity matrix of data samples and selecting appropriate feature vectors. Nevertheless, the original SC method does not have the capability to optimize the task execution order. To ensure the experiment's fairness, we add the designed rolling optimization process to SC.

FCM is a fuzzy clustering algorithm based on an objective function. Similar to SC, when a UAV is assigned a series of tasks, it is impossible to determine the execution order of each task. Hence, we also add the rolling optimization process to FCM. The purpose of comparing with SC and FCM is to prove the accuracy of DP in emergency task scheduling, so that UAVs can complete as many assigned tasks as possible at the minimum cost.

DPGA is proposed to solve the path planning problem of multiple UAVs in a bounded number of regions. In terms of task allocation, this method uses the DP-based algorithm to partition task regions into clusters so as to perform tasks correctly and effectively. In addition, in the task order adjustment phase, DPGA takes the total flight length of the UAV as the fitness function and uses the genetic algorithm (GA) to optimize iterations to find the optimal flight path. Therefore, the purpose of choosing DPGA as the comparison method is to illustrate the effectiveness of the designed rolling optimization mechanism. The specific parameter settings of GA are as follows: population size = 50, crossover rate = 0.8, mutation rate = 0.01, maximum iterations =100, generations = $100 * $ length($U_i$.taskorder).

The UAVs' flight routes generated by the four methods are shown in Figure 5. In Figure 5a, a few task points are too far apart from other task points for the flight routes of $U_1$, $U_4$, and $U_6$, and the positions of the task points performed by other UAVs are not compact. The flight route of $U_1$ in Figure 5b is appropriate, but $U_3$ also has some tasks: it is far away from the task cluster it belongs to. Especially from the route of $U_6$, it can be seen that the assigned task points can be obviously divided into two sub-clusters, which increases the flight time and service cost of the UAV formation. The task allocation results of DPGA and RISE are better than those of SC and FCM, but there are still three task points far away from the task cluster in Figure 5c. Although the task points of $U_6$ are sparse in the UAV route generated by RISE, the closeness between the tasks performed by other UAVs is greater than that of DPGA. This is because we improve the original DP algorithm and select the task points that are far away from the cluster, then reassign these task points to other UAVs, which can reduce the flight distance of UAVs as much as possible and enhance the overall efficiency of UAV formation.

The task scheduling results of the four methods are listed in Table 5. For both general and emergency tasks, 197 tasks have been successfully scheduled by RISE, which is the largest number of tasks completed among the four methods. In comparison to SC, FCM, and DPGA, RISE's running time and flight time of UAV formation are shorter. In addition, the number of SC, FCM, and RISE accesses to the base stations is the same; the only difference is that DPGA is 15. The running time of DPGA is 4.07 s, which is the longest among all methods. Its flight time of UAV formation is 1253.14 s. These results also further prove that the proposed rolling optimization mechanism can not only reasonably optimize the execution order of unscheduled tasks but also have lower time complexity within the premise of maximizing task execution efficiency.

**Table 5.** Comparison results of the four methods.

| Method | Number of Scheduled Tasks | Number of Access to Base Stations | Running Time (s) | Flight Time of UAV Formation (s) |
|---|---|---|---|---|
| SC | 170 | 14 | 0.10 | 1198.22 |
| FCM | 182 | 14 | 0.18 | 1183.24 |
| DPGA | 172 | 15 | 4.07 | 1253.14 |
| RISE | 195 | 14 | 0.06 | 1149.27 |

The comparison results for the number of unscheduled tasks of the four methods are shown in Figure 6. After 200 s, the number of unscheduled tasks of all UAVs tends to surge, which means that emergency tasks arrive in the system and are allocated to each UAV according to the clustering results. However, as shown in Figure 6a, SC, FCM, and RISE all have a clear downward trend after 800 s. The results indicate that numerous tasks are rejected by UAVs at that moment because some constraints in the optimization model (flight distance, resource requirements, etc.) cannot be satisfied. It can be observed

from Figure 6b,d,f that the number of tasks of DPGA, SC, and FCM on $U_2$, $U_4$, and $U_6$ has plummeted, respectively, and many tasks will not be executed, resulting in a low task completion rate. In comparison, RISE rarely has a sudden drop in the number of tasks, which can enable all UAVs to complete as many assigned tasks as possible, thus improving the task completion rate.

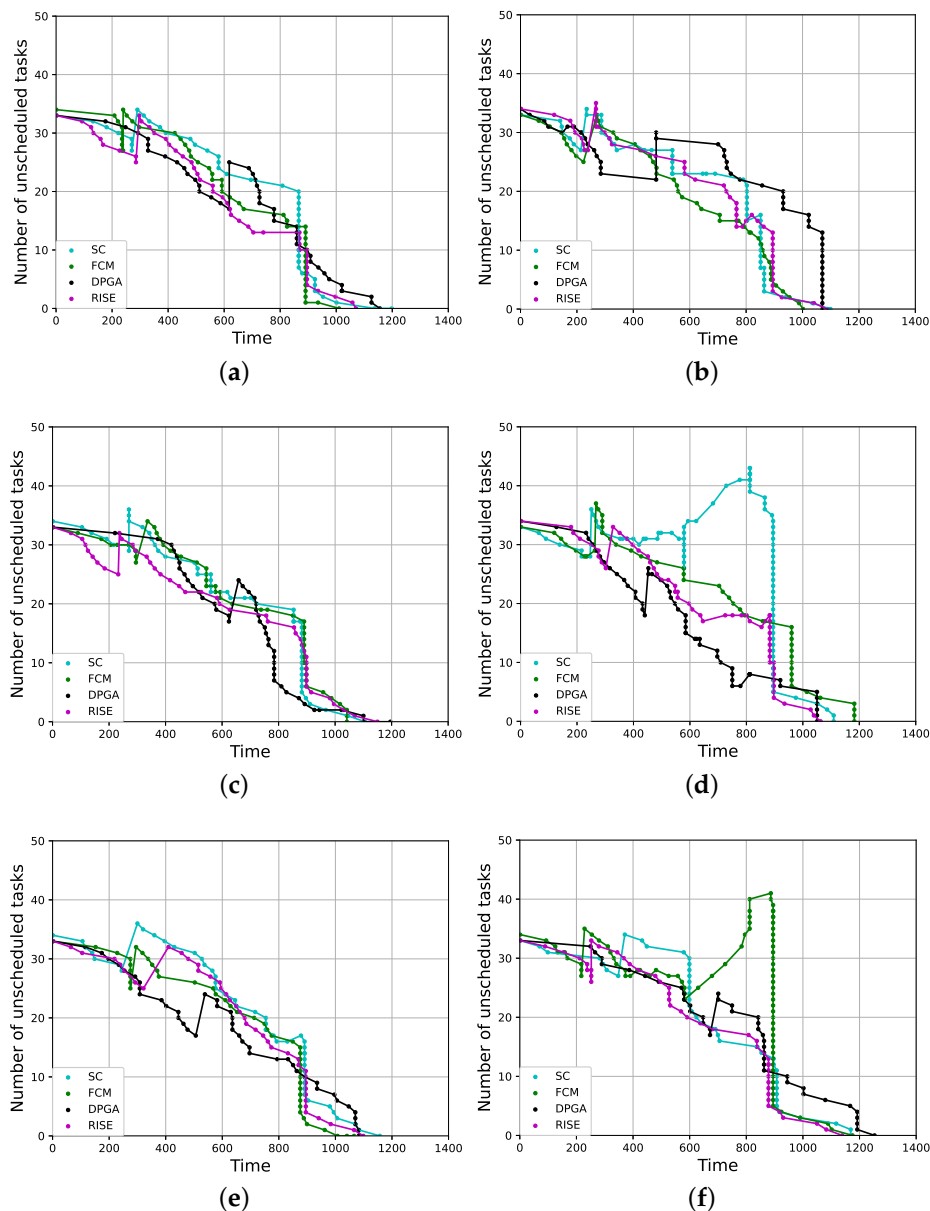

**Figure 6.** Comparison for the number of unscheduled tasks of the four methods: (**a**) $U_1$; (**b**) $U_2$; (**c**) $U_3$; (**d**) $U_4$; (**e**) $U_5$; (**f**) $U_6$.

On this basis, in order to more clearly show the task allocation of the four methods, the number of tasks performed by each UAV is illustrated in Table 6. From Table 6, it can be seen that UAV4 and UAV5 are assigned more tasks than other UAVs for SC; for FCM, $U_2$, and $U_6$ are assigned more tasks than other UAVs; for DPGA, $U_3$, $U_4$, and $U_5$ perform more tasks than $U_1$, $U_2$, and $U_6$. This situation will make UAVs with a small number of tasks end their work in a short time and stay idle, whereas other UAVs cannot meet their maximum mileage due to too many tasks to be performed, and some tasks will not be completed. In order to reduce the flight cost of multiple UAVs, we improved the original DP algorithm and adjusted the number of tasks in each cluster after clustering the tasks

through the proposed balanced allocation mechanism. RISE has the capability to handle outliers and can make the number of task members in each cluster more balanced, further improving the resource utilization of the multi-UAV system.

**Table 6.** The number of tasks performed by each UAV.

| Method<br>UAV ID | SC | FCM | DPGA | RISE |
|---|---|---|---|---|
| $U_1$ | 24 | 24 | 27 | 34 |
| $U_2$ | 25 | 40 | 23 | 30 |
| $U_3$ | 27 | 27 | 34 | 36 |
| $U_4$ | 36 | 27 | 33 | 36 |
| $U_5$ | 33 | 27 | 32 | 32 |
| $U_6$ | 25 | 37 | 23 | 27 |

Because the initial cluster task points are randomly selected for all methods in the clustering phase, there will be certain differences in the results of each run. We run RISE and the three comparison methods 20 times and calculate the average value to reduce the randomness of the experiment and obtain more accurate task scheduling results. The results are shown in Table 7 and Figure 7. In Table 7, we select six indicators to evaluate the performance of all methods, including the maximum number of scheduled tasks, the minimum number of scheduled tasks, the number of scheduled tasks, the number of accesses to base stations, the running time, and the flight time of the UAV formation. It can be seen that RISE shows superior performance: the number of scheduled tasks is 203, and the overall flight time of the formation is 1157 s. The explanation for this phenomenon is that we use an improved DP-based method to cluster the initial task points, and the results can assist the UAVs to obtain more reasonable unscheduled task sequences. Moreover, for each UAV, all the tasks assigned to it are more compact in location space, and it can perform these tasks at the minimum cost within a limited number of resource replenishments. In addition, the comparison results of running time also indicate that the proposed method is superior to SC, FCM, and DPGA in terms of time cost. Furthermore, there are outliers in both SC and FCM but not in DPGA and RISE. For example, among the 20 results of SC, the number of scheduled tasks at one time is the smallest, only 157, which proves that using DP to cluster tasks is more reasonable and stable.

**Table 7.** Comparison results of the four methods after running 20 times.

| Method | Maximum Number of Scheduled Tasks | Minimum Number of Scheduled Tasks | Average Number of Scheduled Tasks | Number of Accesses to Base Stations | Average Running Time (s) | Average Flight Time of UAV Formation (s) |
|---|---|---|---|---|---|---|
| SC | 189 | 157 | 170 | 15 | 0.12 | 1190 |
| FCM | 203 | 172 | 185 | 14 | 0.16 | 1204 |
| DPGA | 185 | 160 | 171 | 14 | 3.85 | 1263 |
| RISE | 214 | 189 | 203 | 14 | 0.05 | 1157 |

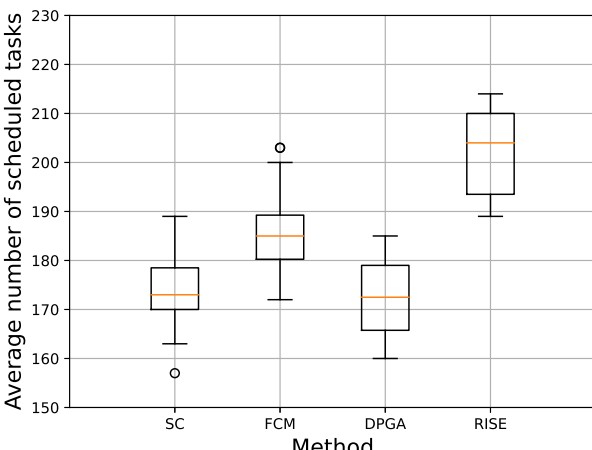

**Figure 7.** Average number of scheduled tasks by the four methods.

*6.2. Extended Experimental Analysis*

6.2.1. RISE Superiority Discussion

In the previous experiments, we selected one instance, including 200 general tasks and 50 emergency tasks, to verify the performance of the proposed method by comparing RISE with other methods. It is a challenge for multi-UAV task scheduling to handle the dynamic changes in the number of tasks arriving at the system. In this section, we conduct a group of experiments with ten instances of each comparison method to further prove the effectiveness of RISE. Instances I1–I5 contain 50 emergency tasks, but their general tasks are different. The number of general tasks generated for these instance ranges is [100, 300] with a step of 50. Different from I1–I5, I6–I10 contain 100 emergency tasks. We also run these instances 20 times and calculate their average values.

Table 8 shows the results of instances I1–I10 for the four methods. The comparative items consist of the number of scheduled tasks, the number of accesses to base stations, and the running time of the algorithm. As can be observed from Table 8, the number of scheduled tasks for RISE is greater than that of SC and FCM in all instances, which indicates that DP is feasible and effective as a task clustering method in our work. Furthermore, as the number of accesses to base stations is fewer than that of SC and FCM, it demonstrates that the proposed task grouping method is reasonable, so that each UAV can perform the assigned task at a lower cost. Compared with DPGA, the number of tasks scheduled by RISE is greater than that of DPGA, except for I7. This means that using rolling optimization to achieve task scheduling can achieve better results than using GA. It can also be seen that, according to the running time of all methods, DPGA is greater than that of SC, FCM, and RISE, and the running time of RISE is less than that of SC and FCM in most instances. The designed task grouping and optimal scheduling method can run at a lower time cost and obtain an optimized task scheduling scheme.

Figure 8 shows the flight time of the UAV formation in each group of instances. From Figure 8, we can see that RISE's overall flight time of the UAV formation is superior to SC and FCM in most instances. Although DPGA is smaller in I1 and I6 than the other three methods, and its average flight time in I6 is only 1014 s, it can be seen from the results in Table 8 that the number of tasks successfully scheduled by DPGA is the fewest, and the running time of DPGA is greater than that of the other methods. In summary, our method achieved excellent performance in solving the task scheduling problem for multiple UAVs.

**Table 8.** Comparison results of ten instances generated by the four methods.

| Method | Instance | Number of General Tasks | Number of Emergency Tasks | Number of Scheduled Tasks | Number of Accesses to Base Station | Running Time |
|--------|----------|------------------------|---------------------------|---------------------------|------------------------------------|--------------|
| SC | I1 | 100 | 50 | $135 \pm 10$ | $14 \pm 1$ | $0.09 \pm 0.02$ |
| | I2 | 150 | 50 | $157 \pm 11$ | $14 \pm 1$ | $0.08 \pm 0.05$ |
| | I3 | 200 | 50 | $170 \pm 19$ | $15 \pm 2$ | $0.12 \pm 0.05$ |
| | I4 | 250 | 50 | $192 \pm 20$ | $15 \pm 2$ | $0.11 \pm 0.05$ |
| | I5 | 300 | 50 | $218 \pm 17$ | $15 \pm 2$ | $0.13 \pm 0.04$ |
| | I6 | 100 | 100 | $160 \pm 9$ | $14 \pm 1$ | $0.04 \pm 0.02$ |
| | I7 | 150 | 100 | $199 \pm 5$ | $15 \pm 1$ | $0.08 \pm 0.03$ |
| | I8 | 200 | 100 | $247 \pm 12$ | $16 \pm 1$ | $0.13 \pm 0.03$ |
| | I9 | 250 | 100 | $272 \pm 21$ | $16 \pm 2$ | $0.15 \pm 0.05$ |
| | I10 | 300 | 100 | $308 \pm 20$ | $16 \pm 2$ | $0.13 \pm 0.08$ |
| FCM | I1 | 100 | 50 | $135 \pm 10$ | $14 \pm 1$ | $0.05 \pm 0.02$ |
| | I2 | 150 | 50 | $163 \pm 7$ | $14 \pm 1$ | $0.10 \pm 0.02$ |
| | I3 | 200 | 50 | $185 \pm 18$ | $14 \pm 2$ | $0.16 \pm 0.05$ |
| | I4 | 250 | 50 | $198 \pm 15$ | $15 \pm 2$ | $0.12 \pm 0.03$ |
| | I5 | 300 | 50 | $235 \pm 13$ | $15 \pm 1$ | $0.12 \pm 0.05$ |
| | I6 | 100 | 100 | $155 \pm 15$ | $14 \pm 1$ | $0.07 \pm 0.02$ |
| | I7 | 150 | 100 | $203 \pm 16$ | $15 \pm 1$ | $0.11 \pm 0.05$ |
| | I8 | 200 | 100 | $250 \pm 10$ | $15 \pm 1$ | $0.07 \pm 0.02$ |
| | I9 | 250 | 100 | $288 \pm 23$ | $16 \pm 2$ | $0.12 \pm 0.06$ |
| | I10 | 300 | 100 | $307 \pm 25$ | $17 \pm 1$ | $0.16 \pm 0.05$ |
| DPGA | I1 | 100 | 50 | $127 \pm 9$ | $13 \pm 1$ | $2.53 \pm 0.50$ |
| | I2 | 150 | 50 | $155 \pm 10$ | $13 \pm 2$ | $3.21 \pm 0.30$ |
| | I3 | 200 | 50 | $171 \pm 14$ | $14 \pm 1$ | $3.85 \pm 0.50$ |
| | I4 | 250 | 50 | $189 \pm 19$ | $15 \pm 2$ | $5.03 \pm 0.20$ |
| | I5 | 300 | 50 | $208 \pm 13$ | $14 \pm 2$ | $5.98 \pm 0.50$ |
| | I6 | 100 | 100 | $155 \pm 15$ | $12 \pm 1$ | $3.11 \pm 0.30$ |
| | I7 | 150 | 100 | $216 \pm 15$ | $14 \pm 1$ | $3.22 \pm 0.50$ |
| | I8 | 200 | 100 | $230 \pm 10$ | $15 \pm 1$ | $3.70 \pm 0.30$ |
| | I9 | 250 | 100 | $258 \pm 18$ | $15 \pm 2$ | $4.46 \pm 0.50$ |
| | I10 | 300 | 100 | $277 \pm 21$ | $15 \pm 2$ | $5.68 \pm 0.20$ |
| RISE | I1 | 100 | 50 | $145 \pm 12$ | $14 \pm 1$ | $0.03 \pm 0.02$ |
| | I2 | 150 | 50 | $182 \pm 8$ | $14 \pm 1$ | $0.03 \pm 0.02$ |
| | I3 | 200 | 50 | $203 \pm 14$ | $14 \pm 2$ | $0.05 \pm 0.03$ |
| | I4 | 250 | 50 | $228 \pm 15$ | $14 \pm 1$ | $0.09 \pm 0.02$ |
| | I5 | 300 | 50 | $257 \pm 10$ | $15 \pm 1$ | $0.11 \pm 0.06$ |
| | I6 | 100 | 100 | $175 \pm 10$ | $14 \pm 2$ | $0.05 \pm 0.02$ |
| | I7 | 150 | 100 | $205 \pm 15$ | $15 \pm 1$ | $0.08 \pm 0.03$ |
| | I8 | 200 | 100 | $256 \pm 22$ | $15 \pm 2$ | $0.10 \pm 0.03$ |
| | I9 | 250 | 100 | $290 \pm 17$ | $15 \pm 1$ | $0.10 \pm 0.05$ |
| | I10 | 300 | 100 | $315 \pm 15$ | $15 \pm 1$ | $0.12 \pm 0.05$ |

6.2.2. RISE Scalability Discussion

The implementation of RISE is significantly impacted by the application scenario's complexity. Therefore, in order to demonstrate RISE's adaptability to the dynamic task environment, we conduct a group of experiments with varying numbers of UAVs and emergency tasks in this section, so as to objectively evaluate the performance and scalability of RISE. In the experiments, the number of general tasks is set to a fixed value $M' = 200$; the number of emergency tasks is within [100, 400] with a step of 100; and the number of UAVs is set to N = 6,9, and 12, respectively. Considering the heterogeneity of the UAVs, the attribute values of newly added UAVs are selected randomly according to the settings in Table 3. We also run each case of RISE 20 times and calculate its average value.

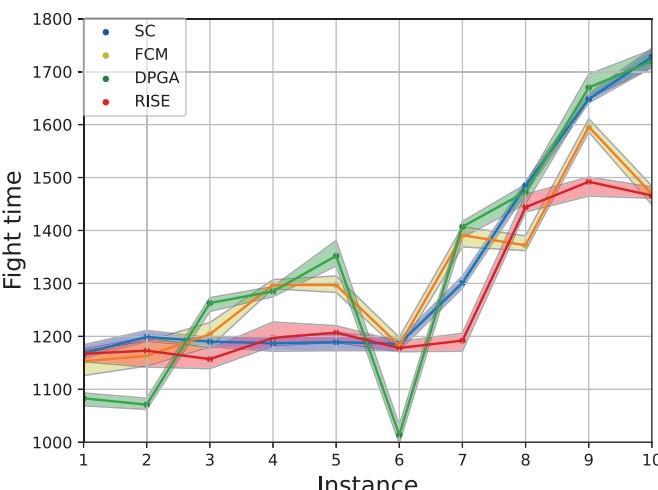

**Figure 8.** The flight time of the UAV formation in each group of instances.

The experimental results are shown in Figure 9 and Table 9. It can be seen from Figure 9 that when $N = 9$ and $N = 12$, the scheduling success rate of RISE is maintained at an excellent level with the increase of emergency tasks. Even if the number of emergency tasks is larger than the number of general tasks, RISE can still generate a reasonable task scheduling scheme so that most tasks are completed before their deadline. RISE has a low scheduling success rate when $N = 6$. This is because the number of emergency tasks is too large for the UAVs to complete the assigned tasks quickly. In addition, the running time in Table 9 further indicates that RISE has an acceptable running efficiency for different cases. In conclusion, RISE has good scalability and superior performance to cope with more complicated task scenarios.

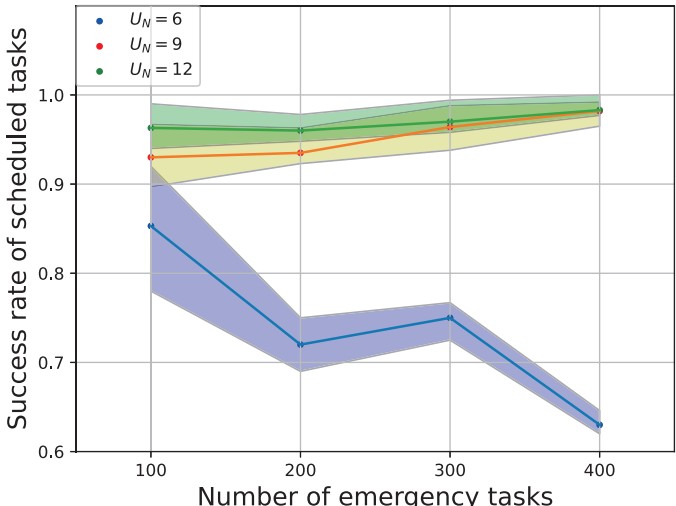

**Figure 9.** The success rate of scheduled tasks.

**Table 9.** Experimental results with different numbers of UAVs.

| Number of UAVs | Number of General Tasks | Number of Emergency Tasks | Number of Scheduled Tasks | Running Time |
|---|---|---|---|---|
| $U_N = 6$ | 200 | 100 | $256 \pm 22$ | $0.10 \pm 0.03$ |
| | 200 | 200 | $288 \pm 12$ | $0.11 \pm 0.05$ |
| | 200 | 300 | $375 \pm 13$ | $0.15 \pm 0.04$ |
| | 200 | 400 | $378 \pm 9$ | $0.22 \pm 0.07$ |
| $U_N = 9$ | 200 | 100 | $279 \pm 11$ | $0.09 \pm 0.05$ |
| | 200 | 200 | $374 \pm 11$ | $0.12 \pm 0.06$ |
| | 200 | 300 | $482 \pm 13$ | $0.19 \pm 0.05$ |
| | 200 | 400 | $589 \pm 10$ | $0.15 \pm 0.07$ |
| $U_N = 12$ | 200 | 100 | $289 \pm 8$ | $0.09 \pm 0.06$ |
| | 200 | 200 | $384 \pm 7$ | $0.05 \pm 0.03$ |
| | 200 | 300 | $485 \pm 12$ | $0.08 \pm 0.04$ |
| | 200 | 400 | $590 \pm 10$ | $0.13 \pm 0.04$ |

## 7. Conclusions

In this paper, we attempt to address the problem of multi-UAV scheduling for emergency tasks. We develop a multi-objective optimization model of task scheduling by taking into account the profit of task completion and the flight cost of UAVs. On this basis, we employ an improved DP algorithm to cluster the tasks and allocate the emergency tasks of the dynamic arrival system to the established clusters, so as to achieve rapid matching between tasks and UAVs. For the unscheduled task sequence of each UAV, a rolling optimization mechanism is proposed to adjust the task scheduling scheme in a real-time manner to obtain the optimal task execution order. The experimental results demonstrate that the improved DP algorithm can obtain more reasonable task clustering results than SC and FCM, which assists UAVs to perform tasks at a lower flight cost. In addition, the proposed rolling optimization mechanism enables UAVs to schedule tasks with higher importance preferentially on the premise of satisfying constraints such as flight distance and resource requirements, so as to maximize the profits of task completion. RISE provides an excellent solution for the dynamic task scheduling problem of multiple UAVs, and it has a shorter running time than the GA-based optimization algorithm. In the future, we will concentrate on the multi-UAV scheduling problem in unknown environments, especially the establishment of a multi-UAV cooperation mechanism under restricted communication conditions.

**Author Contributions:** Conceptualization, B.F. and D.L.; methodology, B.F.; software, B.F. and D.L.; validation, B.F. and D.L.; writing—original draft preparation, B.F.; writing—review and editing, B.F., W.B., X.Z. and M.Z.; visualization, B.F. and D.L.; supervision, W.B. and X.Z.; funding acquisition, X.Z., B.F. and D.L. All authors have read and agreed to the published version of the manuscript.

**Funding:** The authors are grateful for the National Natural Science Foundation of China (61872378); the China Postdoctoral Science Foundation (2020M683723, 2020M673698).

**Institutional Review Board Statement:** Not applicable.

**Informed Consent Statement:** Not applicable.

**Data Availability Statement:** Contact the first/corresponding author please.

**Conflicts of Interest:** The authors declare that they have no conflict of interest.

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
