# Peer review of "RISE: Rolling-Inspired Scheduling for Emergency Tasks by Heterogeneous UAVs"

_drones, doi:10.3390/drones6100310_

Round 1

Reviewer 1 Report

The paper proposes a multi-UAV task scheduling approach based on a task clustering method and a subsequent task ordering.

The paper is generally well-structured and the topic is relevant. However, the description of the state-of-the-art, the newly developed method and the experiments is sometimes unstructed and lacks some important aspects to fully understand the proposed method and the implications of the experiments. Further details below.

The used English is generally readable, but contains some style errors.

The paper provides an extensive state-of-the-art, but fails to discuss the quantitative properties of the solutions. The possible solutions are shortly described and compared regarding very generic properties, of which one is not explained at all: Task type. The resulting quality of solutions and the resources needed to compute them are not mentioned in the discussion.

Line 137 contains a clearly wrong statement, as multi-objective optimization and optimization with constrained are not necessarily best  solved with heuristics. Especially linear objectives and linear constraints may also be solved in polynomial time and therefore exactly.

The description of the method is unstructured as Table 2 clearly shows: The descriptions in the table uses T_i, T_k, B_i, U_i … without introducing them before.

Figure 1: provides an Overview of the algorithms, but it is not clear, what is depicted. The shown relations are not sequences or data flows. Typical outputs to be expected are missing, e.g. task sequences for UAVs.

The assumptions taken regarding the system model are very specific and do not generally fit real-world problems. The used resource model is based on a simple payload model, which is generally not applicable because multiple types of payloads may exist, which create additional constraints for the task allocation. Additionally, UAVs may have different energy storing capabilities, but replenishing them at the base station takes a fixed amount of time. However, the charging time of UAVs typically depend on the battery size. Leaving out height of UAVs is a simplification greatly limiting applicability, and it is unclear what the reasons and consequences of this design decisions are.

A constant energy draw for an UAV is not generally correct. As the energy, majorly depends on weight and the weight scales with payload.

The inter task distance d_{i,j,k} depends on the UAV, but is defined solely on the tasks ...

The meaning and source of the task value v_k and probability pr_k is not defined.

The description of the \eta_{i,j,k} variable is very misleading, as it indicates the allocation of two tasks to an UAV and not the capability of an UAV to fly from one task to another.

Equation 8 is redundant, as it is a special case of equation 7.

Equation 9 constructs an overly complicated logical clock (Lamport Clock).

Equation 10 is not enough to guarantee correct ordering of tasks, as an UAV may have less than M tasks scheduled. In my opinion, this C2 already guarantees sequentiality.

Equations 15 and 16 contains undefined symbols.

Equation 17 is not understandable as the life cycle and type of a task is not described. Additionally, the regularization of Tasks is not described.

In Line 370 European Distance probably means Euclidean Distance.

The design of the evaluation part of the paper cannot convince a reader that the proposed method is generally well suited for the task.

The used algorithms for comparison are either changing the clustering or the task scheduling component, but there is no entirely different algorithm like a pure GA for comparison.

There is no baseline, like a greedy scheduling, where an UAV always flies to the closest task.

It is not clear why the speed of the UAVs was not variated.

The quantitative evaluations omit nearly all relevant parameters like the alpha values generating the final objective as well, as the GA parameters of DPGA (Population size, Mutation Rate, Crossover Rate, Generations ...)

The impact of these parameters is not evaluated, which is problematic as all used algorithms use at least 1 component from the proposed method (either the clustering or the task scheduling)

Table 6 is unnecessary verbose as the task sequence is not informative, only the length is relevant for evaluation.

Section 6.2 is missing stochastic evaluations like standard deviations and stochastic test to verify that the proposed approach indeed is better.

In general, the paper needs additional work to provide a clear method description and sound results. Additional discussion is necessary to clearly depict the use-case fitting to the assumptions.

Reviewer 2 Report

This paper presents the multi-UAV scheduling problem and proposes a method of rolling-inspired scheduling for emergency tasks by heterogeneous UAVs (RISE).The paper is very well structured in terms of method analysis, algorithm implementation and performance evaluation. The content of the paper is well-written, while below suggestions will be helpful for improvement of the paper.

1.The research results can be detailed in the abstract with more sentences.

2.Please provide more explanations of the conclusion from corresponding studies.

3.How do the results support the conclusion? How does the conclusion align with the aim?

Round 2

Reviewer 1 Report

The authors added and clarified many points in the paper. However, many explanations are only comments in the author's reply to my review.

These would benefit the article, if these were also integrated in the paper directly to clearly and transparently describe the assumptions and limitations associated with the scientific work.

Especially the answers to comments: 6, 19,

Regarding Comment 18:

I also think that a greedy scheduling approach will be less efficient, but without an experimental test, this is only a hypothesis. Consequently, it is unclear if the problem to be solved is complex enough for RISE to really provide a benefit. Therefore, I think it is a valid baseline experiment to generate an independent comparison target. Afterwards, the performance can be estimated in a more objective way and the used scenario can be assessed regarding its complexity.
